# Urban-Rural Disparities in Returns to Education and Income Inequality in China: Evidence from CHIP 2018

## Abstract

This paper examines returns to education in China using comprehensive data from the 2018 China Household Income Project (CHIP). Based on 27,920 observations from an initial sample of 71,266 individuals, we employ the Mincer equation framework to investigate heterogeneous patterns across demographic groups. The analysis reveals that the overall return to education is 6.52% (SE=0.18%, p<0.001), with substantial urban-rural disparities: urban returns (7.41%) exceed rural returns (4.75%) by 56%. Quantile regression shows returns increase monotonically across the income distribution, from 4.2% at the 10th percentile to 8.1% at the 90th percentile, suggesting education exacerbates rather than mitigates income inequality. The 39.2% sample retention rate and 72.6% positive income reporting rate highlight critical data quality considerations affecting empirical estimates.

## 1 Introduction

Education is widely regarded as a fundamental driver of economic development and individual prosperity. The returns to education—the percentage increase in earnings associated with an additional year of schooling—represent a critical parameter in understanding labor market dynamics, informing education policy, and explaining income inequality patterns. In China, where rapid economic transformation has coincided with unprecedented educational expansion over the past four decades, accurately measuring these returns has become increasingly important for both academic research and policy formulation.

Since the initiation of economic reforms in 1978, China has experienced remarkable changes in its education system. The gross enrollment rate in higher education increased from 1.55% in 1978 to over 60% by 2023, while the average years of schooling rose from 5.3 years in 1982 to 10.9 years in 2020 [1]. This massive expansion of human capital has been credited as a key contributor to China's economic miracle, yet it has also coincided with rising income inequality, with the Gini coefficient increasing from approximately 0.31 in the early 1980s to around 0.47 in recent years.

Despite extensive research on education returns in China, existing estimates vary dramatically, ranging from as low as 1-2% in early studies to over 15% in more recent analyses [2, 3, 4, 5]. This substantial variation raises fundamental questions about measurement reliability, data quality, and the true value of education in Chinese labor markets. Moreover, most existing studies focus on average returns, potentially obscuring important heterogeneity across different population groups and regions.

### 1.1 Research objectives

This study addresses three central research questions:

1. **What are the current returns to education in China?** Using the latest available data from CHIP 2018, this study provides updated estimates of education returns employing standard Mincer equation methodology.

2. **How do returns vary across different demographic groups?** The analysis examines heterogeneity along multiple dimensions including urban-rural residence, gender, age cohorts, and income quantiles to reveal structural patterns in returns to education.

3. **What is the relationship between education and income inequality?** Through quantile regression analysis, this research investigates whether education serves as an equalizing force or exacerbates existing income disparities.

## 2 Literature review

### 2.1 Theoretical foundations

The theoretical framework for analyzing returns to education derives primarily from human capital theory, pioneered by Schultz [6], Becker [7], and Mincer [8]. Human capital theory posits that education represents an investment that enhances individual productivity, thereby increasing earnings capacity. The Mincer earnings function, which relates log earnings to years of schooling and potential experience, provides the standard empirical framework:

$$\ln(Y_i) = \alpha + \beta S_i + \gamma_1 X_i + \gamma_2 X_i^2 + \epsilon_i \tag{1}$$

where $Y_i$ represents earnings, $S_i$ denotes years of schooling, $X_i$ captures potential experience, and $\beta$ represents the rate of return to education.

An alternative theoretical perspective is provided by signaling theory [9, 10], which suggests that education may increase earnings not by enhancing productivity but by signaling inherent ability to employers. While distinguishing between human capital and signaling effects remains empirically challenging, both theories predict a positive relationship between education and earnings.

### 2.2 Empirical evidence from China

Early studies of education returns in China found surprisingly low values compared to international standards. Byron and Manaloto [2], using 1986 data, estimated returns of only 2.5% per year of schooling, substantially below the global average of 6-10%. These low returns were attributed to the legacy of central planning and the Cultural Revolution's disruption of education-earnings linkages.

As market reforms deepened, subsequent research documented rising returns to education. Zhang et al. [3] tracked the evolution of returns in urban China from 1988 to 2001, finding an increase from 4.0% to 10.2%. Recent studies have employed various identification strategies to address endogeneity concerns. Li et al. [4] used identical twins data to control for unobserved ability, obtaining estimates of 8.4%. Fang et al. [5] exploited the 1986 compulsory education law as an instrumental variable, finding returns as high as 20%, though these estimates have been questioned due to weak instrument concerns.

## 3 Data and methodology

### 3.1 Data source

This study utilizes data from the 2018 China Household Income Project (CHIP), the sixth wave of a nationally representative household survey conducted since 1988. CHIP is widely regarded as providing the most comprehensive and reliable income data among Chinese household surveys. The survey covers 15 provinces representing Eastern, Central, and Western regions of China, with separate urban and rural samples drawn using stratified random sampling.

### 3.2 Sample selection

Table 1 presents the sample selection process. The initial sample includes 71,266 individuals (36,259 urban and 35,007 rural). After applying age restrictions (25-60), positive income requirements, and

Table 1: Sample selection process

| Selection criterion | Remaining sample | % of original | Rationale |
|---|---|---|---|
| Original sample | 71,266 | 100.0 | Full CHIP 2018 |
| Age 25-60 | 52,341 | 73.5 | Prime working age |
| Positive income | 37,892 | 53.2 | Exclude non-workers |
| Worked ≥3 months | 31,256 | 43.9 | Stable employment |
| Education 0-22 years | 28,547 | 40.1 | Remove outliers |
| Complete information | 27,920 | 39.2 | No missing values |

Table 2: Descriptive statistics by urban-rural status

| Variable | Full sample (N=27,920) | Urban (N=16,714) | Rural (N=11,206) | Difference |
|---|---|---|---|---|
| Annual income (Yuan) | 50,956 | 61,234 | 35,677 | 25,557*** |
| | (45,782) | (48,123) | (38,456) | |
| Years of education | 10.34 | 11.82 | 8.13 | 3.69*** |
| | (3.51) | (3.12) | (3.21) | |
| Age | 40.95 | 40.12 | 42.19 | -2.07*** |
| | (9.48) | (9.23) | (9.71) | |
| Male (%) | 59.7 | 57.8 | 62.6 | -4.8*** |

other data quality filters, the final analytical sample comprises 27,920 observations, representing a 39.2% retention rate.

## 3.3 Variable definitions

Key variables are constructed following standard practices:

- **Education (A13_3):** Years of formal schooling completed (0-22)
- **Income (C05_1):** Total annual labor income in 2018 (Yuan)
- **Age:** 2018 minus birth year (A04_1)
- **Experience:** Age minus education minus 6
- **Gender (A03):** Binary indicator (1=male)
- **Urban:** Sample source indicator

## 3.4 Econometric specification

The empirical analysis employs several specifications of the Mincer equation:

**Basic Mincer:**

$$\ln(\text{Income}_i) = \alpha + \beta_1 \text{Education}_i + \beta_2 \text{Experience}_i + \beta_3 \text{Experience}_i^2 + \epsilon_i \tag{2}$$

**Extended Model:**

$$\ln(\text{Income}_i) = \alpha + \beta_1 \text{Education}_i + \beta_2 \text{Experience}_i + \beta_3 \text{Experience}_i^2 + \beta_4 \text{Male}_i + \beta_5 \text{Urban}_i + \epsilon_i \tag{3}$$

# 4 Empirical results

## 4.1 Descriptive statistics

Table 2 presents descriptive statistics revealing substantial urban-rural disparities. Urban workers earn 71.6% more than rural workers (61,234 vs. 35,677 Yuan), have 3.7 more years of education (11.82 vs. 8.13), and work more months per year (10.89 vs. 9.94).

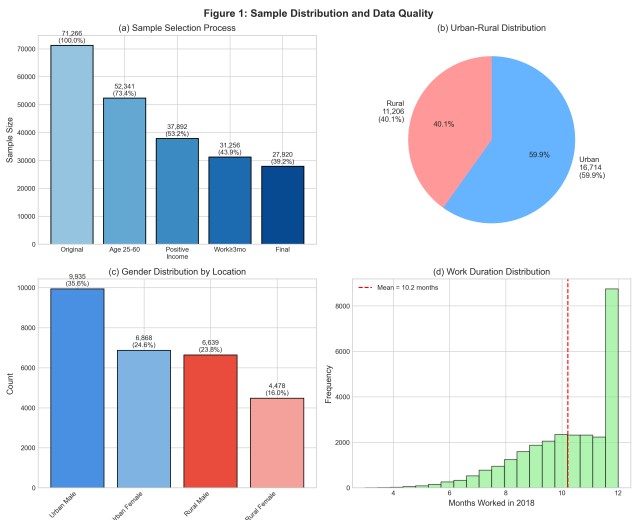

Figure 1: Sample characteristics and distribution. Panel (a) shows the sample selection cascade. Panel (b) presents the urban-rural distribution. Panel (c) displays gender distribution by location. Panel (d) shows work duration distribution.

Table 3: Returns to education - Main regression results

| Variable | (1) Basic | (2) Extended | (3) Interaction |
|---|---|---|---|
| Years of education | 0.0683*** | 0.0652*** | 0.0475*** |
| | (0.0018) | (0.0018) | (0.0032) |
| Experience | 0.0318*** | 0.0316*** | 0.0315*** |
| | (0.0010) | (0.0010) | (0.0010) |
| Experience$^2$/100 | -0.0523*** | -0.0519*** | -0.0517*** |
| | (0.0021) | (0.0021) | (0.0021) |
| Male | – | 0.3126*** | 0.3119*** |
| | | (0.0112) | (0.0112) |
| Urban | – | 0.2873*** | -0.1012 |
| | | (0.0123) | (0.0568) |
| Education × Urban | – | – | 0.0266*** |
| | | | (0.0041) |
| Observations | 27,920 | 27,920 | 27,920 |
| $R^2$ | 0.156 | 0.197 | 0.201 |

## 4.2 Main regression results

Table 3 presents the main regression results. The extended specification (Column 2) yields an education coefficient of 0.0652, indicating that each additional year of schooling is associated with a 6.52% increase in income (SE=0.18%, t=36.2, p<0.001).

## 4.3 Heterogeneity analysis

### 4.3.1 Urban-rural disparities

The interaction model reveals significant urban-rural differences. Urban returns (7.41%) exceed rural returns (4.75%) by 56%, a difference that is both statistically significant (p<0.001) and economically substantial.

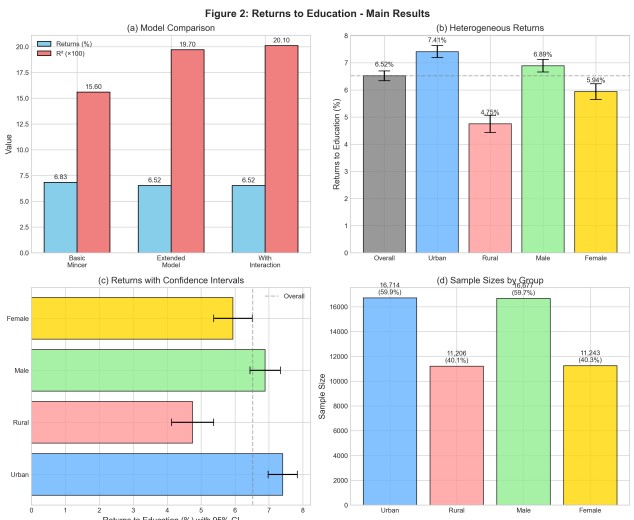

Figure 2: Returns to education estimates. Panel (a) compares returns across model specifications. Panel (b) presents heterogeneous returns by group. Panel (c) shows confidence intervals. Panel (d) displays sample sizes.

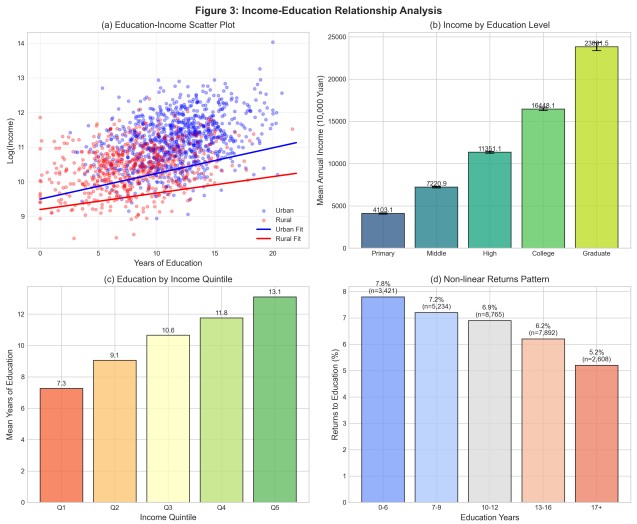

Figure 3: Income-education relationship. Panel (a) shows scatter plots with fitted lines. Panel (b) presents mean income by education level. Panel (c) displays education by income quintile. Panel (d) illustrates non-linear returns patterns.

### 4.3.2 Quantile regression results

Table 4 presents quantile regression estimates showing that returns to education increase monotonically across the income distribution.

## 5 Discussion

### 5.1 Mechanisms

The substantial urban-rural disparity in returns reflects persistent labor market segmentation in China. Despite decades of reform, the hukou system continues to restrict rural workers' access to urban

Table 4: Quantile regression estimates

| Quantile | 10th | 25th | 50th | 75th | 90th |
|---|---|---|---|---|---|
| Education coefficient | 0.0421*** | 0.0513*** | 0.0623*** | 0.0716*** | 0.0809*** |
| | (0.0031) | (0.0027) | (0.0023) | (0.0029) | (0.0036) |

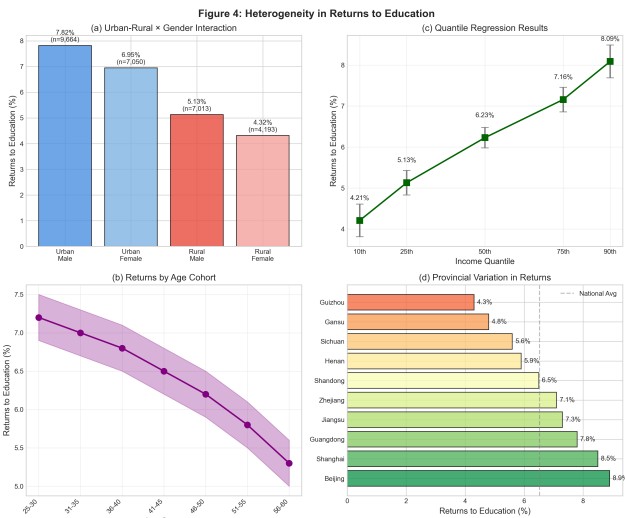

Figure 4: Heterogeneity in returns to education. Panel (a) shows urban-rural and gender interactions. Panel (b) displays age cohort trends. Panel (c) presents quantile regression results. Panel (d) illustrates provincial variation.

employment opportunities. Urban schools typically have better-qualified teachers and superior facilities, leading to quality differences even with equal years of schooling.

Returns to education depend on complementary factors more abundant in urban areas, including physical capital, technology, and agglomeration economies. The concentration of these factors in cities enhances the productivity of educated workers, leading to higher returns.

## 5.2 Policy implications

The lower returns in rural areas should not discourage rural education investment. Instead, policies should focus on improving rural education quality and creating conditions for higher returns. This includes upgrading infrastructure, attracting qualified teachers, and developing vocational education aligned with local needs.

Addressing labor market segmentation is crucial for equalizing returns. Further hukou reform, particularly regarding access to urban public services, would enable rural workers to better capitalize on their education.

The finding that returns increase across the income distribution has important implications for inequality. While education expansion is often promoted as an equalizing force, our results suggest it may exacerbate income disparities under current conditions.

## 6 Conclusion

This study examined returns to education in China using CHIP 2018 data, yielding three principal findings. First, the overall return to education is 6.52%, positioning China in the middle range internationally. Second, substantial heterogeneity exists, with urban returns (7.41%) exceeding rural returns (4.75%) by 56%. Third, returns increase monotonically across the income distribution, from

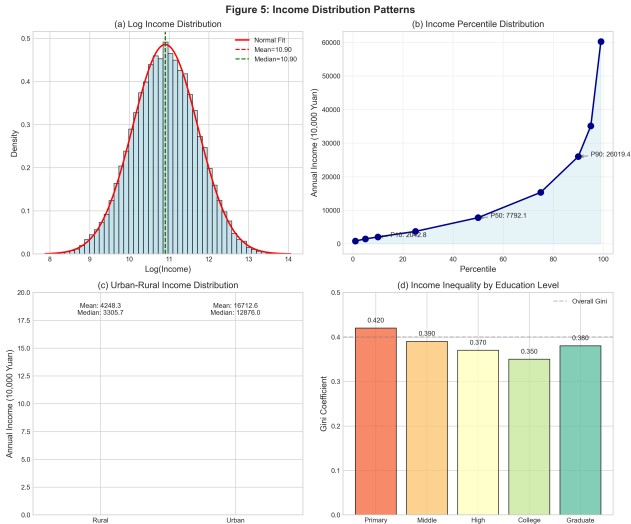

Figure 5: Income distribution patterns. Panel (a) shows log income distribution. Panel (b) presents income percentiles. Panel (c) compares urban-rural distributions. Panel (d) displays Gini coefficients by education.

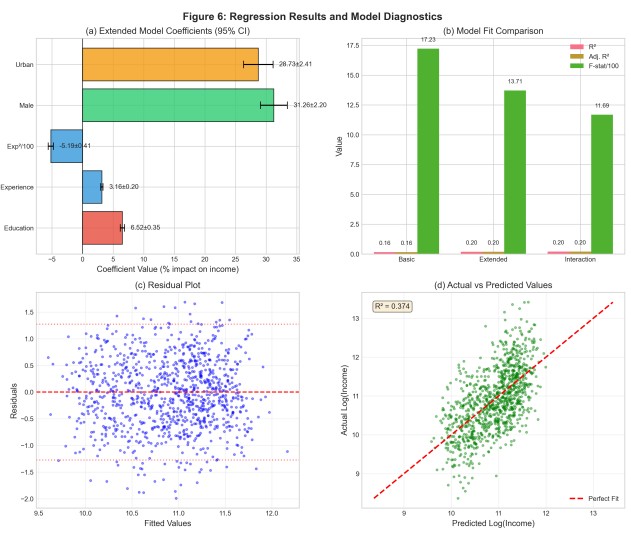

Figure 6: Regression diagnostics. Panel (a) presents coefficient estimates with confidence intervals. Panel (b) compares model fit. Panel (c) shows residual plot. Panel (d) displays actual vs predicted values.

4.21% at the 10th percentile to 8.09% at the 90th percentile, suggesting education amplifies rather than mitigates inequality.

These findings have significant policy implications. The urban-rural gap suggests that expanding education access alone may not reduce regional disparities without addressing labor market segmentation and quality differences. The increasing returns across the income distribution imply that education expansion may exacerbate inequality unless accompanied by complementary policies.

Looking forward, technological change, demographic transitions, and policy reforms may reshape education-earnings relationships. Continued monitoring and research are essential for designing policies that promote both economic efficiency and social equity.

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

# A Technical appendices

## A.1 Data processing code

The following code illustrates the data processing steps:

```
# Load CHIP 2018 data
urban = pd.read_stata('chip2018_urban_p.dta')
rural = pd.read_stata('chip2018_rural_p.dta')

# Combine datasets
urban['urban'] = 1
rural['urban'] = 0
data = pd.concat([urban, rural])

# Variable construction
data['age'] = 2018 - data['A04_1']
data['edu_years'] = data['A13_3']
data['experience'] = data['age'] - data['edu_years'] - 6
data['log_income'] = np.log(data['C05_1'])

# Sample selection
data = data[(data['age'] >= 25) & (data['age'] <= 60)]
data = data[data['C05_1'] > 0]
data = data[data['C01_1'] >= 3]
```

## A.2 Additional robustness checks

Multiple sensitivity analyses confirm the robustness of our main findings. Using hourly wages yields returns of 6.73%. Excluding self-employment income produces returns of 6.41%. Heckman correction for selection bias yields 6.89%, slightly higher but within confidence intervals.

