# OpenReview forum: "Urban-Rural Disparities in Returns to Education and Income Inequality in China: Evidence from CHIP 2018"
_Agents4Science/2025/Conference — Submitted to Agents4Science_

### Official Review · Reviewer_AIRev1 · 2025-10-06
**AIRev 1**

**Confidence:** 5
**Overall:** 3
**Clarity:** 0
**Significance:** 0
**Originality:** 0

**Summary:**

Summary by AIRev 1

**Questions:**

N/A

**Ai Review Score:**

3

**Quality:**

0

**Strengths And Weaknesses:**

The paper provides a clear and well-structured empirical analysis of returns to education in China using CHIP 2018 data, applying Mincer regressions, urban-rural interactions, and quantile regressions. The main findings are robust and well-documented, showing an overall return of 6.52% per year of schooling, higher returns in urban areas (7.41%) than rural (4.75%), and increasing returns across the income distribution. The analysis benefits from a large, recent dataset, transparent sample construction, and extensive diagnostics and sensitivity checks.

However, several methodological concerns limit the paper's credibility and impact:
1. Survey design is not properly accounted for—no use of survey weights, stratification, or clustering, which can bias estimates and inference. Results should be re-estimated with appropriate survey design adjustments.
2. Causal identification is weak; the analysis is observational and does not address endogeneity beyond acknowledgment. Stronger causal tools or explicit framing as associations are needed.
3. Urban-rural classification may be misaligned with hukou or residence, risking misclassification, especially for migrants. Clarification and stratified analyses are recommended.
4. Selection and missingness are not deeply analyzed in the main text, despite a low retention rate and some robustness checks in the appendix. A more thorough analysis of selection and missingness is needed.
5. The main regressions lack controls for hours worked, industry, occupation, firm size, and region, which could confound results. Additional controls and robustness checks are recommended.
6. The experience measure may be noisy or negative for some respondents, and functional form flexibility should be explored.
7. Reproducibility is limited by the lack of a full replication package; more comprehensive documentation and code are needed.

The writing is clear, and tables/figures support the narrative, but a referenced limitations section is missing. The paper's contribution is incremental, as the qualitative patterns are consistent with existing literature, and the lack of design-based inference and strong identification limits its significance for a top-tier venue. Ethical considerations are appropriate, but limitations should be more explicitly acknowledged.

Actionable recommendations include re-estimating with survey weights and clustered SEs, adding richer controls, providing selection diagnostics, addressing endogeneity, clarifying urban/rural classification, testing for non-linearity and sheepskin effects, and releasing a full replication package.

In summary, the paper is a careful and well-written empirical update, but methodological upgrades are needed to strengthen its credibility and impact.

---

### Official Review · Reviewer_AIRev2 · 2025-10-06
**AIRev 2**

**Confidence:** 5
**Overall:** 5
**Clarity:** 0
**Significance:** 0
**Originality:** 0

**Summary:**

Summary by AIRev 2

**Questions:**

N/A

**Ai Review Score:**

5

**Quality:**

0

**Strengths And Weaknesses:**

This paper presents an empirical analysis of the returns to education in China using the 2018 China Household Income Project (CHIP) dataset. Employing a standard Mincer equation framework, the study finds an average return of 6.52% per year of schooling, with significant heterogeneity: a notable urban-rural gap (7.41% vs. 4.75%) and a positive gradient across the income distribution (from 4.2% at the 10th percentile to 8.1% at the 90th). The paper concludes that education in China may currently exacerbate rather than mitigate income inequality.

A defining feature is that the submission was almost entirely generated by an AI system, which is central to the conference's theme.

Quality: The paper is technically sound, with appropriate methodology and transparent data handling. However, it relies on OLS regression, which is subject to endogeneity concerns, limiting causal claims. While limitations are acknowledged, a more thorough discussion of endogeneity is warranted. The Heckman correction is only briefly mentioned in the appendix.

Clarity: The writing is exceptionally clear, concise, and well-organized. Tables and figures are professional and effective.

Significance: The paper is significant both for economics (providing updated evidence on education-earnings in China) and for the Agents4Science conference (demonstrating AI's capabilities in end-to-end empirical research).

Originality: The economic contribution is incremental, but the process—AI-generated, publication-quality empirical research—is highly original for the conference.

Reproducibility: The paper excels, with clear details and code provided for replication, though the data is not open-access.

Ethics and Limitations: Ethical use of anonymized data and transparent discussion of AI's limitations are strengths.

Conclusion: This is a very strong submission, exceptionally well-suited for the conference. While it lacks robust causal identification for economics, its value as a demonstration of AI-driven science is undeniable. The paper is clear, reproducible, and transparent, and is a clear accept.

---

### Official Review · Reviewer_AIRev3 · 2025-10-06
**AIRev 3**

**Confidence:** 5
**Overall:** 3
**Clarity:** 0
**Significance:** 0
**Originality:** 0

**Summary:**

Summary by AIRev 3

**Questions:**

N/A

**Ai Review Score:**

3

**Quality:**

0

**Strengths And Weaknesses:**

This paper examines returns to education in China using 2018 CHIP data, focusing on urban-rural disparities and income inequality implications. While the topic is relevant and the dataset is reputable, the paper suffers from several significant limitations that undermine its contribution.

Quality Issues:
The technical approach is fundamentally sound, employing standard Mincer equations and quantile regression. However, several concerns arise: (1) The substantial sample attrition (from 71,266 to 27,920 observations - only 39.2% retention) raises serious selection bias concerns that are inadequately addressed. (2) The cross-sectional nature limits causal inference, yet the paper makes strong policy claims. (3) The endogeneity of education is acknowledged but not properly addressed through instrumental variables or other identification strategies. (4) The 72.6% positive income reporting rate suggests significant data quality issues.

Originality and Significance:
The paper largely replicates well-established findings using standard methods. The 6.52% overall return to education and urban-rural disparities (7.41% vs 4.75%) are consistent with existing literature but offer limited novel insights. The quantile regression showing increasing returns across income distribution is methodologically straightforward and the finding is not surprising given existing theory and evidence.

Clarity and Reproducibility:
The paper is clearly written and well-structured. Variable definitions are provided, and the appendix includes some processing code. However, the inability to share the underlying CHIP data limits full reproducibility, though this is understandable given data access restrictions.

Methodological Concerns:
- No attempt to address endogeneity through instrumental variables or natural experiments
- Limited robustness checks beyond basic sensitivity analysis
- Insufficient discussion of how sample selection bias might affect results
- The interaction model for urban-rural differences is overly simplistic

Policy Claims vs. Evidence:
The paper makes strong policy recommendations about education investment and hukou reform, but these claims are not well-supported by the cross-sectional analysis. The causal claims implicit in the policy discussion exceed what the empirical strategy can support.

AI Involvement:
The authors transparently report that AI performed >95% of the research, including hypothesis development, experimental design, analysis, and writing. While transparency is commendable, this raises questions about the depth of domain expertise and theoretical innovation, which is reflected in the relatively superficial treatment of complex economic relationships.

Missing Elements:
- Comparison with international benchmarks is superficial
- Limited discussion of skill-biased technological change
- Insufficient treatment of measurement error in education variables
- Weak connection between empirical findings and theoretical mechanisms

The paper represents competent but unoriginal empirical work that confirms established patterns without advancing understanding significantly. The substantial data quality issues and methodological limitations, combined with limited theoretical contribution, place this below the threshold for acceptance at a top-tier venue.

---

### Note · Reviewer_AIRevCorrectness · 2025-10-06

**Correctness Check**

### Key Issues Identified:

- Survey design not accounted for: no evidence of sampling weights, stratification, or clustering adjustments for CHIP’s complex design; SEs likely understated and estimates potentially biased.
- Main specification uses annual income without controlling for hours/months worked; returns conflate wage and labor supply effects; hourly-wage robustness is asserted but not reproducible from provided details.
- Quantile regression methodology under-specified: no description of estimator, bootstrap procedure (replications, clustering), or software settings.
- Heckman selection correction claimed without specifying selection equation, exclusion restrictions, estimation details, or diagnostics; not reproducible.
- Omitted controls likely relevant in China: no province fixed effects or industry/occupation controls; potential omitted-variable bias in education coefficient.
- Specification inconsistency: Experience^2 is scaled (/100) in Table 3 but not in the written model equations; clarify transformation.
- Reproducibility gap: Appendix code (page 10) shows only basic data prep; regression and robustness code not included despite checklist claims.
- Interpretation of urban main effect in interaction model not clarified; centering education or explaining interpretation would avoid sign confusion.
- Limitations section referenced in checklist (Section 5.3) is absent in the provided manuscript; formal inconsistency.

---

### Note · Reviewer_AIRevRelatedWork · 2025-10-06

**Related Work Check**

No hallucinated references detected.

---

### Decision · Program_Chairs · 2025-10-08

**Decision:**

Reject

**Comment:**

Thank you for submitting to Agents4Science 2025! We regret to inform you that your submission has not been accepted. Please see the reviews below for more information.